# Unsymmetrical Trifluoromethyl Methoxyphenyl β-Diketones: Effect of the Position of Methoxy Group and Coordination at Cu(II) on Biological Activity

**DOI:** 10.3390/molecules26216466

**Published:** 2021-10-26

**Authors:** Liliya A. Khamidullina, Igor S. Puzyrev, Gennady L. Burygin, Pavel V. Dorovatovskii, Yan V. Zubavichus, Anna V. Mitrofanova, Victor N. Khrustalev, Tatiana V. Timofeeva, Pavel A. Slepukhin, Polina D. Tobysheva, Alexander V. Pestov, Euro Solari, Alexander G. Tskhovrebov, Valentine G. Nenajdenko

**Affiliations:** 1Postovsky Institute of Organic Synthesis, Ural Branch of the Russian Academy of Sciences, 22 Sofia Kovalevskaya Street, 620137 Ekaterinburg, Russia; khamidullina@ios.uran.ru (L.A.K.); slepukhin@ios.uran.ru (P.A.S.); pestov@ios.uran.ru (A.V.P.); 2Institute of Natural Sciences and Mathematics, Ural Federal University, 19 Mira Street, 620002 Ekaterinburg, Russia; albus.p.wulfric.b.d@gmail.com; 3Institute of Biochemistry and Physiology of Plants and Microorganisms, Russian Academy of Sciences, 13 Prospekt Entuziastov, 410049 Saratov, Russia; burygingl@gmail.com; 4NRC “Kurchatov Institute”, 1 Acad. Kurchatov Sq., 123182 Moscow, Russia; paulgemini@mail.ru; 5Boreskov Institute of Catalysis, Siberian Branch of the Russian Academy of Sciences, 5 Lavrentiev Ave., 630090 Novosibirsk, Russia; yvz@catalysis.ru; 6Department of Inorganic Chemistry, Peoples’ Friendship University of Russia (RUDN University), 6 Miklukho-Maklay Street, 117198 Moscow, Russia; chemistann@gmail.com (A.V.M.); vnkhrustalev@gmail.com (V.N.K.); 7Zelinsky Institute of Organic Chemistry RAS, 47 Leninsky Prosp., 119991 Moscow, Russia; 8Department of Chemistry, New Mexico Highlands University, Las Vegas, NM 87701, USA; tvtimofeeva@nmhu.edu; 9Institute of Chemical Sciences and Engineering, Ecole Polytechnique Fédérale de Lausanne (EPFL), CH 1015 Lausanne, Switzerland; euro.solari@epfl.ch; 10Chemistry Department, Lomonosov Moscow State University, 1/3 Leninskie Gory, 119991 Moscow, Russia

**Keywords:** copper(II), diketonates, cytotoxic activity, antimicrobial activity, anticancer agents

## Abstract

Copper(II) complexes with 1,1,1-trifluoro-4-(4-methoxyphenyl)butan-2,4-dione (H**L1**) were synthesized and characterized by elemental analysis, FT-IR spectroscopy, and single crystal X-ray diffraction. The biological properties of H**L1** and *cis*-[Cu(**L1**)_2_(DMSO)] (**3**) were examined against Gram-positive and Gram-negative bacteria and opportunistic unicellular fungi. The cytotoxicity was estimated towards the HeLa and Vero cell lines. Complex **3** demonstrated antibacterial activity towards *S. aureus* comparable to that of streptomycin, lower antifungal activity than the ligand H**L1** and moderate cytotoxicity. The bioactivity was compared with the activity of compounds of similar structures. The effect of changing the position of the methoxy group at the aromatic ring in the ligand moiety of the complexes on their antimicrobial and cytotoxic activity was explored. We propose that complex **3** has lower bioavailability and reduced bioactivity than expected due to strong intermolecular contacts. In addition, molecular docking studies provided theoretical information on the interactions of tested compounds with ribonucleotide reductase subunit R2, as well as the chaperones Hsp70 and Hsp90, which are important biomolecular targets for antitumor and antimicrobial drug search and design. The obtained results revealed that the complexes displayed enhanced affinity over organic ligands. Taken together, the copper(II) complexes with the trifluoromethyl methoxyphenyl-substituted β-diketones could be considered as promising anticancer agents with antibacterial properties.

## 1. Introduction

Cancer is the second leading cause of death globally (just after cardiovascular diseases) according to a surveillance report compiled by WHO [1]. The major cancer treatment options include surgery, radiation therapy, and chemotherapy. Anti-tumor drugs used for chemotherapy often induce undesired side effects [2] with damage to the rapidly self-renewing cells of the hematopoietic and immunocompetent organs being dominant. Neutropenia is a frequently occurring sequela of drug therapy, which enhances the probability of the occurrence of life-threatening bloodstream infections requiring stationary treatment in a hospital, including antibiotic courses [3,4]. The development of neutropenia is promoted by various factors, such as concomitant immunodeficiency (associated with the tumor, immunosuppressive therapy applied, or repeated surgical interventions) and chronic deceases [5]. Therefore, many lethal cases with immune-compromised cancer patients are due to hospital-acquired infections [6].

Furthermore, infections with microbial pathogens themselves remain among the five most frequent causes of death [7]. According to recent investigations, the list of most common causative agents for nosocomial infections includes (in descending order of frequency) *Clostridium difficile, Staphylococcus aureus, Klebsiella pneumoniae/Klebsiella oxytoca*, and *Escherichia coli*, which are responsible for pneumonia and surgical site and gastrointestinal infections [8]. It is of note that *Staphylococcus aureus* is one of the predominant causes of bloodstream infections [4]. A common strategy for the prevention and treatment of hospital-acquired and cancer therapy–related infections implies the use of a broad spectrum antibiotic to address as many potential bacterial pathogens as possible, which gives rise to the emergence of antimicrobial resistance [3]. The ever-growing resistivity of pathogen microorganisms against commonly applied antibiotics, degrading the efficiency of decease prevention and treatment measures, is a global challenge that health care organizations have to meet urgently before the world enters the post-antibiotic era [7,9,10].

Many researchers stress the necessity to combine an anti-cancer chemotherapy with dedicated measures preventing nosocomial infections [5,11]. This requires a preliminary assessment of potential side effects of preventive courses of antibiotics to ensure that anticipated benefits exceed potential risks [12]. The use of anti-cancer drugs that simultaneously manifest antimicrobial activity seems very promising since they complement the therapeutic treatment of the basic disease with targeted protection of a patient against potential associated infections, avoiding increased drug burden. The development of such medicines thus pursues both goals: cancer therapy and the prevention of bacterial infections.

Screening of the biological activity of newly synthesized metal complexes is a prospective tool in the search for novel highly efficient drugs [13,14,15,16] that demonstrate activity patterns drastically different from standard ones. For instance, copper-based complexes are known to act as proteasome inhibitors that could be targeted to human cancer cells due to their elevated proteasome activity and the high concentration of copper therein [17,18,19,20]. Copper complexes are considered to be a potent alternative to platinum agents owing to the lower toxicity of endogenous metals for normal cells with respect to tumor cells, as well as novel modes of action including the inhibition of alternative targets and the induc-tion of oxidative stress resulting from Cu(II) redox transformations [21,22,23,24,25,26].

The bioessentiality of copper for the majority of living organisms has a dual character of either a vital or a lethal trace element depending on the specific biochemical process involved [27,28]. Antibacterial properties of copper are known from ancient times and are still in use [29,30]. Several mechanisms of that action have been suggested [28]. Therefore, the search for Cu-based metal complexes manifesting anti-tumor and antimicrobial activities simultaneously is an urgent need.

Many β-diketones demonstrate diverse types of biological activity, such as cytotoxic, antioxidant, antiangiogenic, anti-tumor, antibacterial, and fungicidal activities [31,32,33,34,35,36,37]. Metal complexes with β-diketones as ligands are even more interesting since the introduction of metal ions makes it possible to overcome the emerging resistance of bacteria against pristine drugs [30]. Furthermore, the complexes can act as biomimetics of the superoxide dismutase that plays the key role in the antioxidant protection of both healthy and tumor cells by catalyzing the dismutation of superoxide into oxygen and hydrogen peroxide [38]. They are also capable of either promoting metal ion metabolism or inhibiting the migration and proliferation of HeLa cells [16,39]. Some of the complexes have been selected for clinical trials that are either planned or underway now [15,26].

Recently, we have synthesized and investigated biological properties of trifluoromethyl- and methoxyphenyl-containing copper(II) β-diketonates [40]. We found that the complexes demonstrated multiple antibacterial and antifungal activity. In this paper, we present synthesis and bioactivity studies of new copper(II) β-diketonates, as well as a comparison of their structures and properties with those reported earlier. Aiming to evaluate the effect of methoxy group position and the binding of Cu(II) ions on biological properties, antibacterial activity of the copper(II) β-diketonate with the methoxy group at the para-position of the ring was estimated and compared with the activity of copper(II) β-diketonate with the methoxy group at the ortho-position. For assumptions of the activity origin and perspective of the compounds’ uses as flexible chemotherapeutic agents, we included the assessment of the cytotoxic properties of both newly synthesized and previously described structurally isomeric compounds, as well as molecular modeling. These copper complexes are built with the participation of two ligands which have iden-tical donor cores but different positions of methoxy groups at the aromatic ring. This structural modification, which does not lead to a significant change in the donor activity of the diketone fragment, results in the formation of complexes of distinct spatial structures and can change selectivity for biological targets, the pose of substrates in active sites, bioavailability, and activity.

## 2. Results and Discussion

### 2.1. Synthesis and Characterization

Ligand H**L1** was synthesized using similar procedure to that described earlier [40]. A reaction of its sodium salt with Cu(OOCCH_3_)_2_·H_2_O with a copper-to-ligand molar ratio equal to 1:2 in water-methanol medium at room temperature led to a green precipitate. After recrystallization of the product from DMF, green single crystals of (N,N-dimethylformamide-O)-bis[(p-methoxybenzoyl)-trifluoroacetonato-O,O’]copper(II) [Cu(**L1**)_2_(DMF)] (**1**) in cis-configuration were obtained in 84% yield (Figure 1).

After recrystallization of complex **1** from methanol, the configuration of the copper bis(β-diketonate) moiety changed from cis to trans. The resultant trans-complex *trans*-[Cu(**L1**)_2_] (**2**) (79% yield) did not include any solvent molecule. Finally, after recrystallization from DMSO which is a strong complexing agent, *cis*-[Cu(**L1**)_2_(DMSO)] (**3**), including five-coordinated copper ions, was obtained in 82% yield. The process was repeated several times and we concluded that either of the complexes can be successfully reproduced in good yield. We did not form complexes incorporating six-coordinated copper ions. Thus, the metal-to-ligand relation in the obtained metal complexes was 1:2. Copper complexes of three different structures were obtained according to specific solvent chosen for the crystallization.

The IR spectrum of the free ligand was registered for comparison to the complexes isolated. Characteristic absorptions in the 3150 to 2830 cm^–1^ region were observed corresponding to ν(C–H) [41]. The ligand existing in the keto-enol form had characteristic bands at 1506 cm^–1^ and 1586 cm^–1^ assigned to ν(C=C), coupled with ν(C=O) [41]. The band at 1506 cm^–1^ appeared in the spectra of the complexes at 1506, 1507 and 1505 cm^–1^. On complexation with copper(II), instead of the intense broad band at 1586 cm^–1^ the four distinct bands emerged. These characteristic bands were located at 1540, 1568, 1589 and 1613 cm^–1^ for complex **1**, at 1543, 1567, 1586 and 1611 cm^–1^ for the complex **2**, and at 1538, 1568, 1588 and 1613 cm^–1^ for complex **3**. In the spectrum of complex **1,** a sharp band at 1666 cm^−1^ belonging to the carbonyl stretch of DMF coordinated to the copper was redshifted relative to its position in free DMF (1675 cm^−1^) [42]. Also, for complexes **1**–**3**, the symmetric CF_3_ stretching vibrations were observed at 1148, 1144, 1147 cm^–1^, and the asymmetric CF_3_ stretching vibrations were observed at 1315, 1318, 1312 cm^–1^, respectively [41,43]. These absorption bands appeared in the IR spectrum of the free ligand at 1130 and 1312 cm^–1^.

### 2.2. Structural Studies

The ligand H**L1** and complexes **1**–**3** were analyzed by single-crystal X-ray crystallography. Figure 1 shows the ellipsoid plots of the H**L1**. The crystallographic data and selected structural parameters are summarized in Appendix A, respectively.

The ligand H**L1** crystallized in the monoclinic space group P2_1_/c. The molecule H**L1** corresponded to the enol tautomeric form. The C1–O1 bond 1.3111(12) Å) was longer than C3–O2 (1.2588(12) Å), whereas the C1–C2 (1.3969(13) Å) and C2–C3 bonds (1.3965(13) Å) were essentially of the same length. There was a typical very strong enolic intramolecular hydrogen bond in the structure, with the O2···O1 distance being 2.4955(12) Å; and the O2–H2–O1 angle as 154.3(16)°. The CF_3_ group is fully ordered, similarly to what we observed earlier [44]. The torsion angle O1–C1–C5–C6 was equal to 7.03(13)°, the major contribution to the deviation from the planarity of the whole molecule came from the rotation around the C1–C5 bond.

The molecules packed in the crystal lattices via weak CH···F and CH···O intermolecular interactions (Appendix A). The hydrogen bond parameters are listed in Appendix A).

The compounds **1** and **3** crystallizes in the monoclinic space group P2_1_/c. The five-coordinated copper atoms in both complexes adopted the same coordination geometries, which is a distorted square pyramidal [45] geometry with two chelating ligands in the cis-configuration, forming its basal plane. Oxygen atoms belonging to respective solvent molecules provided apical coordination to the Cu center. The Cu1–O5 distance in **1** (2.339(3) Å) corresponded to a typical single bond. The dihedral angle between the planes of the chelate rings was 11.08(14)°. The O-atoms of the diketonate ligands lay in the basal plane (deviation from the basal plane <0.02 Å), the Cu-atom deviated from the mean plane of oxygen atoms by 0.11 Å. In the chelate rings, the Cu–O distances were 1.926–1.937 Å, and the C–O distances were 1.252–1.275 Å. Two ligands in the complex **1** were not equivalent, which is related to the presence of the C23–H23A···O4 intramolecular hydrogen bond formed by the DMF molecule. The dihedral angles between the mean planes of the aryl and chelate rings were 1.34(14)° and 3.69(12)°, the ArC–C and CF_3_C–C distances were also slightly different (1.422–1.434 Å and 1.362–1.370 Å, respectively) in two distinct ligand molecules. One of the CF_3_-group (at the C13 position) was disordered over two positions, with the occupancy factors 0.85/0.15.

In complex **3**, the axial Cu–O distance (2.2946(15) Å) conformed to a typical single bond, as in the case of complex **1**. The dihedral angle between the planes of the chelate rings was 26.04(04)°. The oxygen atoms of the diketonate ligands deviated from the basal plane by less than 0.02 Å, the copper atom deviated from the mean plane of the oxygen atoms by 0.14 Å. The bond angles at the basal plane insignificantly deviates from 90°. The basal Cu–O distances were 1.9286–1.9496 Å, the C–O distances were 1.261–1.275 Å. As was the case with **1**, the ligands in complex **3** were not equivalent, owing to presence of the C21–H21···O5 intramolecular hydrogen bond. The dihedral angles between the mean planes of the aryl and chelate rings were 9.93(6)° and 4.00(6)°, and the ArC–C and CF_3_C–C distances were 1.427–1.434 Å and 1.370–1.372 Å, respectively.

Of note is the paramount impact of the oxygen atoms of methoxy groups on the crystal packing, despite the fact that they do not directly participate in the formation of classic Cu–O bonds within molecules. Specifically, the O6 atom interacts with the copper atom of the adjacent molecule, thereby zigzag crystal packing is observed (Figure 2) despite the intermolecular Cu–O distance (2.818(4) Å) being slightly longer than these in polymeric complexes. Importantly, such interaction becomes possible precisely due to the para-position of the methoxy group in H**L1,** in contrast to the analogous ligand having a methoxy group in the ortho-position and forming discrete complexes [40]. The mentioned interaction is more intense in **3** than in **1,** with the Cu1–O3 distance being equal to 2.7495(16) Å.

The crystal packing of complexes **1** and **3** is governed by several short contacts, which may be classified as non-classical hydrogen bonds, and parallel-displaced π−π stacking interactions. The hydrogen bonds parameters are listed in Appendix A. Both phenylene moieties are involved into the intermolecular π−π stacking interactions with the phenylene moieties of adjacent molecules. The molecules are thus linked by π−π stacking into dimers (centroid-centroid distances of 3.955(3) Å and 3.8865(14) Å for **1** and **3**, respectively) (Appendix A).

Complex **2** crystallizes in the triclinic space group P-1 with two crystallographically independent molecules **A** and **B** in the unit cell. The coordination of the copper atom is a slightly distorted square planar with an inversion center at the copper atom, making the two ligands crystallographically identical. The negative charge is delocalized between two oxygen atoms, O1 and O2. In molecule **A**, the longer C2–O1 (1.292(3) Å) bond is closer to a single bond, i.e., it is of a slightly more enolate character than C4–O2 (1.287(3) Å). Herewith, the C2–C3 bond is shorter (1.387(3) Å), than that of C3–C4 (1.440(3) Å), which is in accordance with the enolate character of C2–O1 bonds. The dihedral angle between the mean planes of the aryl rings and chelated cycles was 8.31(6)°.

The molecules of complex **2** in the crystal lattice are held together via intermolecular interactions (Appendix A). The hydrogen bond parameters are listed in Appendix A. In the crystal, molecules are also connected by weak intermolecular interactions between phenyl rings belonging to adjacent molecules (centroid-centroid distance is 3,8313(18) Å).

### 2.3. Cytotoxicity Studies

Correlation of the biological activity of the test organic and coordination compounds with their composition and structure is of interest in the context of the rational design of new promising drugs. In order to evaluate the effect of the inclusion of copper(II) in organic molecules on their activity, cellular studies for H**L1** and *cis*-[Cu(**L1**)(_2_(DMSO)] (**3**) were performed. Complex **3** was chosen for the biological experiments due to the fact that DMSO was used for dissolution as it is a non-toxic solvent that does not affect the development of microorganisms and animal cell lines in the working concentration range. Additionally, for the establishment of the influence of the soft modification of the ligand structure we for the first time tested the previously reported ligand H**L0** and complex *cis*-[Cu(**L0**)_2_(DMSO)_2_] (Figure 3), which have similar structures to those described here [40]. It should be noted that the fine-tuning of ligand provides the means to affect an interaction with specific targets. We suppose the affinity-based biochemical approach to be more advantageous than that used in many studies, pointing to a synergistic metal plus ligand effect.

Of 31 cancer incidences in women, the fourth most frequent type is cervical carcinoma [46]. Hence, the human cervical epithelioid carcinoma cell line was chosen as a tumorigenic. The african green monkey kidney cell line was used as the normal (non-cancerous) line, since it is one of the most common mammalian continuous cell lines.

An MTT assay was used to evaluate the cytotoxic activity of the test compounds (as we performed it earlier) [45,47,48,49]. Representative dose–dependent curves are presented in Figure 4, and maximal inhibition values are summarized in Appendix A. The ligands H**L0** and H**L1** had low activity against both cancer and normal cells. The complexes exhibited moderate activity for the more sensitive HeLa line and low activity for the Vero line. *cis*-[Cu(**L0**)_2_(DMSO)_2_] showed higher activity against HeLa cells (IC_50_ = 86.6 ± 12.1 μM) than *cis*-[Cu(**L1**)_2_(DMSO)] (IC_50_ = 119.9 ± 15.1 μM). In our experiment, the IC_50_ values of cisplatin and carboplatin were equal to 21.2 ± 1.4 and 421.5 ± 7.9 μM, respectively. Therefore, the cytotoxicity of the studied complexes against the HeLa cell line was slightly lower than that of cisplatin, but higher than that of carboplatin.

Since the ligands participating in the binding of copper(II) have slightly different structure features, less reactivity and bioavailability are apparently the limiting factors for **3**. It is probable that the six-coordinated complex *cis*-[Cu(**L0**)_2_(DMSO)_2_] is stable enough to shuttle the metal to the cell without irreversible interactions with physiological entities, while the lability of DMSO co-ligands make it possible for copper to interact with the binding site once the complex has reached the target substrate. Also, it is known that the hydrophilic groups, i.e., DMSO, should contribute to enhanced solubility and better availability [50]. The minor bioavailability of **3** can presumably be explained by the competitive tendency of link formation between copper of one molecule and methoxy oxygen of another, as apparent in the crystal structure (vide supra). Moreover, this tendency can cause the incapability of copper ions to interact with biotargets at the molecular level.

### 2.4. Antimicrobial Activity

According to data from the WHO, the design of new antibiotics is mostly focused in improvements of existing classes [51]. The main drawback of this method is a high risk of cross-resistance. A strategy to overcome this sort of resistance is to find novel chemical structures with new targets and new modes of action. The development of metallopharmaceuticals can represent an innovative approach in search for effective antibacterials.

In accordance with the above, the antibacterial activity of the compounds towards microorganisms, including the most common pathogens causing nosocomial infections (*S. aureus*, *E. coli*), were examined, along with their effects on opportunistic pathogenic fungi (*C. albicans*) invading the bloodstream of immunocompromised patients, giving rise to life-threatening infections. Phytopathogenic bacteria *P. atrosepticum* exhibiting close phylogenetic affinity to *E. coli* and other human pathogenic bacteria belonging to *Entero**bacteria* which were proven using comparative genomic analyses [52], were also included in the study as additional Gram-negative species.

The preliminary screening performed by the disc diffusion method (data are not given) revealed that the tested compounds were effective against the microorganisms. As observed in the experiments, the antibacterial activity of the ligand and complex towards some strains compares with those for commercial antibacterial agents (ampicillin, tetracycline, kanamycin). The compounds produced inhibition zones with diameters comparable to those of the positive controls (Appendix A). Copper(II) acetate was also tested over the concentration range from 1 mM to 100 mM. The experiments showed that the copper(II) ions were inactive towards the used strains. To further determine the activity of the compounds, the agar microdilution method was conducted using the procedure described in Section 4. A dissimilar sensitivity of the mentioned pathogens was observed. The test compounds were found to be active against both Gram-positive and Gram-negative bacteria at the concentration range of 64–512 µg·mL^−1^ (Table 1). A comparative analysis of the MIC values indicated that the complex overall exhibited similar antibacterial activity with the parent ligand. Ligand H**L1** demonstrated stronger inhibiting properties than H**L0** (Appendix A) [40]. However, the incorporation of copper(II) unexpectedly did not result in an increase of activity. Once again, these observations suggest that low bioavailability was the limiting factor for **3**. Nevertheless, the best inhibition activity was recorded towards *S. aureus* ATCC 25923 by complex **3,** which showed a comparable result with streptomycin (31 µg·mL^−1^, 0.02 mM) MIC levels [53]. Of note is that complex **3** had similar or higher activity than those of other Cu(II) β-diketonates (Appendix A) [53,54].

The cytotoxicity of complex **3** against Vero cells was compared with the MIC values for microbes. The complex demonstrated stronger activity towards *S. aureus* and *P. atrosepticum* 34-1/1 than towards the other bacteria and eukaryotic cells. At concentrations ranging from 100 to 200 μM, complete growth inhibition of the sensitive bacteria was observed (Table 1), but only about a 50% inhibition of the viability of Vero cells was detected. Thus, complex **3** is a promising candidate as an anti-staphylococci drug. Further in vivo studies are needed to estimate the potential of the complex in animal models.

In contrast to the effect towards bacteria, the antifungal activity of complex **3** was found to be low. The lower MIC value could be expected for *cis*-[Cu**L1**_2_(DMSO)] as compared with *cis*-[Cu(**L0**)_2_(DMSO)_2_], since ligand H**L1** exhibited a lower MIC value than H**L0**. Conversely, the in vitro activity against *C. albicans* of ligand H**L1** surpassed that of the tested complex (as opposed to H**L0** and related complexes), with an MIC value of 128 µg·mL^–1^. These findings may again be explained by the limited complex **3** bioavailability which probably shows itself as a decreased penetration rate into the cell.

### 2.5. Stability in Simulated Biological Fluids

In order to evaluate the biopersistence of the studied copper diketonates, the stability of complex **3** in a simulated biological fluid (SBF) and an aqueous solution of human serum albumin (HSA) at body temperature was spectrophotometrically determined. The composition of a SBF solution with an ion concentration nearly equal to that of human blood plasma and a pH of 7.4, close to that of a human body [55], is given in Table 2. The results are shown in Figure 5. UV-visible spectra indicated that the addition of Cu(II) ions to solutions of HSA/SBF did not result in appearance of new absorption bands. Upon the addition of H**L1** to HSA/SBF solutions at a concentration of 3·10^–5^ M, an absorption shoulder appeared. The HSA/SBF solutions containing complex **3** at the concentration of 1.5·10^−5^ M displayed more intense absorbance bands. The intensity of new bands with maxima at 320 and 330 nm for HSA and SBF, respectively, increased as the complex was added. The linearity of the obtained concentration dependences (Figure 5, inset plots) demonstrates that the complex remained intact in such environments. Similar to other copper complexes [56], there are probably interactions between the metal complex and albumin, which is responsible for drug transportation.

### 2.6. Docking Study

Cancer, bacterial and yeast cells possess similar properties, such as high rates of proliferation, quick spreading within the host organism, and the capacity to cause disease following a latent period, despite the associated diseases falling into totally distinct categories of medical disorders [57]. DNA double strand breaks must be quickly repaired to guarantee cell survival in stress conditions. In most cells, there are special compounds which express when cells are stressed: for example, such proteins as ubiquitin, chaperones Hsp70 and Hsp90. The heat shock protein (HSP) family is directly involved in DNA repair and serves the abovementioned purpose [58]. Some of the major functions of HSPs are the folding and stabilization of misfolded proteins and prevention of their aggregation. Cancer, bacterial and fungal cells depend on the concentration of these proteins during stress [59,60,61,62]. Post-translational modifications involved in DNA repair are usually accompanied by the subsequent ubiquitin-dependent degradation pathway via proteasomes. However, as a response to DNA damage, mammalian Hsp90α is phosphorylated at the Thr-7 residue and acts as a DNA repair enzyme, preventing degradation of the nucleic acid molecule [58]. Interestingly, Hsp90α interacts with DNA within sites of damage. According to their function, Hsp70 and Hsp90 chaperones stabilize components of ribonucleotide reductase (RNR), which occupies a crucial place in the control of DNA synthesis owing to the catalytic function of this enzyme in the reduction of ribonucleotides to deoxyribonucleotides. Consequently, RNR is an important biomolecular target in the molecular modeling of drugs inhibiting the replication of cancer cells, yeasts, and bacteria [62,63,64]. In order to ensure the fast proliferation and optimal fidelity of DNA replication, cells need to keep appropriate RNA production levels for the maintenance of a high nucleoside triphosphates (NTPs) concentration. RNR bears a large subunit (α or R1) which performs a regulating function, and a small subunit (β or R2) with a unique tyrosyl radical stabilized by an adjacent binuclear iron center, essential for the catalysis of the nucleotide reduction [63,65]. Specific inhibitors of ribonucleotide reductase, excluding de novo DNA synthesis in cancer, bacterial and yeast cells, can be used as therapeutic agents [65]. Moreover, the inhibition of chaperones may reflect on decreases of RNR levels, so the chaperones Hsp70 and Hsp90 may present successful targets for antitumor and antimicrobial drugs as well [62]. The efficacy of target inhibition depends on the access of inhibitors to the essential fragments: the diferric center and tyrosyl residue of RNR and N-terminal nucleotide-binding domains (NBD) of HSPs which are highly conserved across species [66,67,68].

Down-regulating properties of copper complexes can be due to various mechanisms of action and inhibitions of different biotargets [23,26], including RNR [69,70] and HSPs [71]. Due to the coexistence of copper in two oxidation states in living organisms, this metal can induce oxidative stress, which in turn can result in cell death [72]. It should be pointed out that cancerous tissues are more sensitive to copper than normal ones, since copper plays a central role in their growth and metabolism [26]. Hence, the development of new copper-based agents may provide an avenue for the treatment of infectious and cancer diseases.

GOLD is one of the available molecular docking software programs for rational drug design research. It provides results with good correlations with experimental data for the study of metal complexes [73]. Consequently, the molecular docking calculations for predictions of the binding affinity between the target and studied compounds were performed using GOLD. We incorporated calculations of both the trans- and cis-configurations of copper complexes without co-ligand solvents in axial positions since the determination of unique configurations in living cells is complicated. In the docking study, we included complex **2**, previously reported *trans*-[Cu(**L0**)_2_] [40], and the hypothetical complexes *cis*-[Cu(**L1**)_2_] and *cis*-[Cu(**L0**)_2_] (Figure 6).

The resulting protein-ligand complexes were analyzed via the GoldScore and ChemScore functions, which reveal the binding energy of the complexes. The values of the scoring functions for copper complexes was higher than those for organic ligands (Table 3), in general agreement with stronger downregulating activity of metal complexes in vitro. The docked configurations of copper complexes in comparison with the parent ligands in the corresponding binding sites are presented in Appendix A (for GoldScore function). Figure 7, Figure 8 and Figure 9 and Appendix A show selected amino acids and a partial reproduction of a hydrogen-bonding pattern (GoldScore). The organic ligands and metal complexes were located in close proximity to the essential fragments of the targets. In particular, the compounds were arranged within the binding site of RNR nearby the diferric cofactor, necessary for the catalytic activity of RNR (Appendix A). The ligand H**L1** bonded with Glu335 via the oxygen atom of the methoxy group (Figure 7a). The complexes *cis*-[Cu(**L1**)_2_] (Figure 7b) and *cis*-[Cu(**L0**)_2_] (Appendix A) interacted with either or both Ser264 and Cys271. The bond between the complex *cis*-[Cu(**L1**)_2_] and Cys271 was realized due to the suitable position of one of methoxy groups. As depicted in Figure 7c, the oxygen atoms of the methoxy group and the diketo moiety of *trans*-[Cu(**L1**)_2_] were involve in hydrogen bonding to Phe237 and Arg331 residues, respectively. The compounds were deeply embedded into the pocket of the Hsp90 (Appendix A). It was observed that the binding of both *cis*-complexes was stabilized by H-bonds with Asn106 (Figure 8b and Appendix A), while that of *trans*-[Cu(**L0**)_2_] was stabilized by an H-bond with Asn51 (Appendix A). The compounds were placed inside the hydrophilic NBD of Hsp70 (Appendix A) involved in the pocket–substrate interactions required in HSP-catalyzed conformational reactions [74]. The ligands H**L1** and H**L0** were both observed to interact with Hsp70 by forming hydrogen bonds with Lys271 (Figure 9a and Appendix A), while the complexes *cis*-[Cu(**L1**)_2_], *trans*-[Cu(**L1**)_2_] and *cis*-[Cu(**L0**)_2_] bound with Hsp70 through H-bonds with Arg272 (Figure 9b,c and Appendix A).

Therefore, it can assume that the activity of the studied compounds is associated with the inhibition of the abovementioned targets. However, other possible modes of action should not be excluded include oxidative stress caused by redox transitions of copper, proteasome and Topo I inhibition, DNA targeting and ER stress. It should be noted that the ability to simultaneously inhibit various targets is probably an effective way to overcome the possible development of pathogen resistance.

## 3. Conclusions

In conclusion, we have synthesized and characterized for the first time Cu(II) complexes based on trifluoromethyl- and methoxyphenyl-containing β-diketones. The complexes *cis*-[Cu(**L1**)_2_(DMSO)] (**3**) (H**L1** containing a para-methoxyphenyl group) and *cis*-[Cu(**L0**)_2_(DMSO)_2_] (H**L0** containing an ortho-methoxyphenyl group) with structurally isomeric ligands were tested, along with the starting ligands, regarding their cytotoxicity in cellular assays against cancer and normal cell lines. The parent ligands demonstrated low activity against eukaryotic cells, while the complexes showed different activity towards cancer cells and low activity towards normal cells. The toxic activity appears to be dependent on the metal presence, as well as on the position of the methoxy group in the aromatic ring. Differences in the activities of the complexes were probably due to differences in their bioavailability, which are related to the peculiarities of their structure. The antibacterial data in vitro showed that tested complex **3** exhibited similar antibacterial activity with the parent ligand and demonstrated an MIC value towards *S. aureus* comparable to that of streptomycin. The study of antifungal properties in vitro revealed that complex **3** showed lower antifungal activity than ligand H**L1**. Docking studies generally pointed to the enhanced affinity of metal complexes binding to the target sites over the organic ligands and the findings agree with the effect towards eukaryotic cells. Unexpectedly, the reduced activity of complex **3** may be explained by its lower biological availability by comparison to the complex *cis*-[Cu(**L0**)_2_(DMSO)_2_]. At the same time, its distinct behavior against the eukaryotic cells and bacteria/fungi may also be related to the nature and specific defense mechanisms of these bioobjects. In conclusion, the studied complexes demonstrate their potential as new metallo-anticancer agents with antibacterial properties. Taking into consideration the diverse biological activity coming with the insignificant structural differences of ligands and metal complexes, we believe that data on their crystallographic features hold significance in the interpretation of their properties. In particular, strong intermolecular interactions presenting in the solid and resulting in chain structure formation apparently lead to reduced inhibiting effects. Consequently, ligand selection with surgical precision is a key to success in the development of new effective metal-based drugs, and research into the properties of extremely similar complexes is an important rung on the ladder of rational design of metallopharmaceuticals.

## 4. Materials and Methods

### 4.1. General Remarks

Commercially available reagents and solvents used in this study were of pure grade and used without further purification unless otherwise specified. Human albumin (Alburex, 200 g/L, solution for infusion) was obtained from CSL Behring AG (Switzerland). Cisplatin-LANS (0.5 mg/mL, solution for infusion) was obtained from Veropharm Ltd. (Russia). Carboplatin-RONC (10 mg/mL, solution for infusion) was obtained from the FSBI National Medical Research Center of Oncology named after N.N. Blokhin of the Ministry of Health of Russia. FT-IR spectra of the compounds were recorded in the 4000–400 cm^−1^ region using a Spectrum Two (Perkin Elmer, Waltham, WA, USA). ^1^H, ^13^C and ^19^F NMR spectra were obtained on a Bruker Avance DRX-500. TMS and CFCl_3_ were used as internal references. Microanalyses (C, H, N) were conducted using a Perkin Elmer PE 2400 elemental analyzer.

### 4.2. Crystal Structure Determination

Single crystal X-ray diffraction data for H**L1** were collected on Bruker APEX-II CCD diffractometer (λ(MoKα)-radiation, graphite monochromator, ω and φ scan mode) and corrected for absorption using the SADABS program. Single crystal X-ray diffraction data for **1** were collected on an Xcalibur-3 automated four-circle κ-geometry diffractometer with a CCD detector by a standard procedure (graphite monochromated Mo Kα (*λ* = 0.71073 Å) radiation, ω-scanning with a step of 1° at T = 295(2) K) by applying the CrysAlis CCD Software system (Oxford Diffraction), Version 1.171.39.38a. Data reduction was performed by the same programs. The analytical absorption correction by the multifaceted crystal model was applied [75]. Sets of diffraction reflections for **2** and **3** were collected at the “BELOK” beamline of the Kurchatov synchrotron radiation source (NRC, Kurchatov Institute) using a Rayonix SX165 CCD detector (*λ* = 0.96990 Å, *ϕ*-scanning mode with a step of 1.0° at T = 100(2) K, 720 frames at two different crystal orientations). Data processing was performed with the *iMOSFLM* code from the CCP4 suite [76]. Scaling of intensities and semi-empirical absorption correction were performed using the *Scala* program [77]. For details, see Appendix A. The structures H**L1**, **2** and **3** were determined by direct methods and refined by the full-matrix least squares technique on *F*^2^ with anisotropic displacement parameters for non-hydrogen atoms. The hydrogen atom of the hydroxy group was localized in the difference-Fourier map and refined isotropically with fixed displacement parameters (*U*_iso_(H) = 1.5*U*_eq_(O)). The other hydrogen atoms were placed in calculated positions and refined within the riding model with fixed isotropic displacement parameters (*U*_iso_(H) = 1.5*U*_eq_(C) for the CH_3_-groups and 1.2*U*_eq_(C) for the other groups). All calculations were carried out using the SHELXTL program [78]. Structure **1** was solved by direct methods and refined by the full-matrix least squares technique on *F*^2^ in SHELXL-97 [79], including anisotropic displacement parameters for all non-H atoms. H-atoms at C-H bonds were placed in calculated positions and refined in the riding model in the isotropic approximation with correlated thermal parameters. The final geometrical parameters were obtained, and the figures were drawn using the programs OLEX2 [80] and PLATON [81]. CCDC: 1861631, 1861887, 1861635, and 1861636 contain the supplementary crystallographic data for H**L1**, **1**, **2**, and **3**, respectively. These data can be obtained free of charge via http://www.ccdc.cam.ac.uk/conts/retrieving.html (Accessed on: 20 October 2021), from the Cambridge Crystallographic Data Centre, 12 Union Road, Cambridge CB2 1EZ, UK; fax: (+44) 1223-336-033; or through e-mail to deposit@ccdc.cam.ac.uk.

### 4.3. Synthetic Part

#### 4.3.1. Synthesis of H**L1**

Ligand 1,1,1-trifluoro-4-(4-methoxyphenyl)butan-2,4-dione (H**L1**) was synthesized according to the procedure described below. A solution of 4′-methoxyacetophenone (15.0 g, 100 mmol) and ethyl trifluoroacetate (14.3 mL, 120 mmol) in 40 mL of dry tetrahydrofuran in the presence of LiH (2.0 g, 250 mmol) was heated under reflux for 5 h. After removal of the solvent using rotary evaporation, the crude product was triturated with 40 mL of acetic acid in 160 mL of water. The product extracted from the aqueous solution with chloroform was washed by water three times, dried with anhydrous Na_2_SO_4_, and recrystallized from chloroform. M.p. 60–62 °C. Yield: 18.7 g (76%.) Anal. calc. (%) for C_11_H_9_O_3_F_3_ (246.19 g·mol^−1^): C 53.67, H 3.68, F 23.15. Found: C 53.90, H 3.80, F 23.21. IR data (ATR, cm^−1^): *ν* 3120, 2850, 1586, 1506, 1312, 1254, 1192, 1166, 1130, 1109, 1068, 1019, 915, 843, 814, 791. ^1^H NMR of H**L1** (CHCl_3_, δ, ppm, *J*/Hz): 3.81 (s, 3H, CH_3_); 6.10 (s, 1H, CH); 6.97 (d, 2H, ArH, *J* = 8.8); 7.84 (d, 2H, ArH, *J* = 8.8). ^19^F NMR of H**L1** (CHCl_3_, δ, ppm, C_6_F_6_): 88.03 (s, CF_3_). ^1^H NMR of H**L1** (DMSO_d6_, δ, ppm, *J*/Hz): 3.89 (s, 3H, CH_3_); 6.97 (s, 1H, CH); 7.11 (d, 2H, ArH, *J* = 8.7); 8.15 (d, 2H, ArH, *J* = 8.9). ^19^F NMR of H**L1** (DMSO_d6_, δ, ppm): 87.74 (s, CF_3_). The single crystal of H**L1** suitable for the X-ray crystallographic study was obtained by slow evaporation of its chloroform solution.

#### 4.3.2. Synthesis of Complex **1**

Complex **1** was prepared as described below. To a stirred solution of H**L1** (0.98 g, 4 mmol) and NaOH (0.16 g, 4 mmol) in MeOH (20 mL) an H_2_O solution (20 mL) of Cu(OOCCH_3_)_2_·H_2_O (0.40 g, 2 mmol) was added slowly. This reaction mixture was stirred at room temperature for 24 h and the formation of green precipitate was observed. Next, the product was filtered off, washed with water, and dried under reduced pressure. Slow evaporation in air of the DMF solution of the powder gave green block-shaped single crystals suitable for X-ray analysis. Yield: 1.05 g (84%). Anal. calc. (%) for C_25_H_23_CuF_6_NO_7_ (627.00 g·mol^−1^): C, 47.89; H, 3.70; Cu, 10.14; F, 18.18; N, 2.23. Found: C, 47.71; H, 3.62; Cu, 10.23; F, 18.24; N, 2.05. IR data (ATR, cm^−1^): *ν* 2938, 2845, 1666, 1613, 1589, 1568, 1540, 1506, 1383, 1324, 1294, 1266, 1246, 1198, 1174, 1148, 1129, 1118, 1095, 785.

#### 4.3.3. Synthesis of Complex **2**

Complex **1** (0.80 g) was dissolved in methanol and kept for slow evaporation in air. As a result, dark green needle-shaped single crystals of **2** were obtained. Yield: 0.56 g (79%). Anal. calc. (%) for C_22_H_16_CuF_6_O_6_ (553.01 g·mol^−1^): C, 47.71; H, 2.91; Cu, 11.47; F, 20.58. Found: C, 47.65; H, 2.74; Cu, 11.55; F, 20.70. IR data (ATR, cm^−1^): *ν* 3012, 2839, 1611, 1586, 1567, 1543, 1507, 1318, 1296, 1270, 1249, 1173, 1144, 1129, 1119, 1073, 1025, 943, 846, 787.

#### 4.3.4. Synthesis of Complex **3**

To obtain complex **3**, 0.40 g of **2** was dissolved in DMSO, and the obtained solution was kept for slow evaporation in air. Green block-shaped single crystals of **3** were obtained. Yield: 0.37 g (82%). Anal. calc. (%) for C_24_H_22_CuF_6_O_7_S (631.03 g·mol^−1^): C, 45.61; H, 3.51; Cu, 10.05; F, 18.04; S, 5.07. Found: C, 45.49; H, 3.38; Cu, 10.14; F, 17.92; S, 5.20. IR data (ATR, cm^−1^): *ν* 2974, 2844, 1613, 1588, 1568, 1538, 1505, 1312, 1289, 1265, 1246, 1201, 1171, 1147, 1127, 1118, 1069, 1019, 946, 785.

### 4.4. In Vitro Antimicrobial Activity

In vitro antibacterial activity of the test compounds was preliminarily evaluated by the disc diffusion method on a panel of four human pathogenic bacterial species, which included three Gram-positive bacteria, *Staphylococcus aureus* ATCC 25923, *Staphylococcus aureus* ATCC 29213, and *Bacillus subtilis* ATCC 6633; one Gram-negative bacteria *Escherichia coli* ATCC 25922, and two Gram-negative phytopathogenic bacteria *Pectobacterium atrosepticum* RCAM 01724, and *Pectobacterium atrosepticum* 34-1/1 (isolated from *Solanum tuberosum* L. cv. “Bodenkraft”) [82]. All the bacteria were obtained from the Russian National Collection of Industrial Microorganisms except for *P. atrosepticum* RCAM 01724, which was obtained from the Russian Collection of Agriculture Microorganisms. All the isolates were maintained on nutrient meat peptone agar at 4 °C for the described experiments.

The inoculums were prepared by direct suspension of an overnight culture of bacterial strains in sterile distilled water. The suspensions were diluted to a turbidity equivalent to a 1 McFarland standard (3·10^8^ cfu·mL^−1^) [83]. Using a sterile glass spreader, bacterial cultures (100 mL) were used to spread a bacterial lawn on nutrient agar plates. The compounds for the disk diffusion method were dissolved in methanol to a final concentration of 10 mg·mL^–1^, blank paper discs (0.6 cm diameter) were impregnated with the obtained solutions (loading volume 10 μL), dried, and then applied to the surface of inoculated plates. The plates were incubated at 37 °C for 18–24 h, then were examined and the diameters of the inhibition zones were measured. A negative control was prepared by using methanol. Ampicillin, tetracycline and kanamycin (10 μg, Research Center of Pharmacotherapy, Saint Petersburg, Russia) were used as positive references in preliminary experiments using the disk diffusion method.

The agar microdilution method was used for the determination of the minimum inhibitory concentrations (MICs) of test compounds according to the protocol of the Clinical and Laboratory Standards Institute [84]. DMSO served as a diluent for all the compounds; the final concentrations did not exceed 5% and were not found to be harmful to any of the cell lines. Meat peptone agar was used as the nutrient medium. Inoculum suspensions were prepared to give a final concentration of approximately 10^5^–10^6^ cfu·mL^−1^. The suspensions were used to inoculate the agar plates supplemented with the test compounds. Petri dishes contained the nutrient medium with 2-fold serial dilutions of test compounds to give final concentrations ranging from 0.008 to 1.024 mg/mL. After setting of the agar, the plates were inoculated with standardized inocula, and further incubated at 37 °C (27 °C for phytopathogens). The MICs were read visually after 24 h and (48 h for phytopathogens) as the lowest concentration that consistently prevented visible growth in all three trials. MICs were determined twice and in triplicate.

The opportunistic fungal pathogen *Candida albicans* ATCC 10231 was included in the experiments. The strain was obtained from the Russian Collection of Agriculture Microorganisms and maintained on wort agar media at 4 °C. Prior to the experiments, *C. albicans* colonies were streaked onto agar plates and incubated at 27 °C for 48 h. The activity of the compounds against *C. albicans* was tested as mentioned above.

### 4.5. In Vitro Cytotoxic Activity

Cytotoxic activity was evaluated using the colorimetric MTT assay previously described [85]. Vero (African green monkey kidney epithelial cells) [86] and HeLa (human cervical epithelioid carcinoma) [87] cells were provided by the Lab of Nanobiotechnology staff, IBPPM RAS (Saratov, Russia). Cell lines were cultured as adherent monolayers at 37 °C with a 5% CO_2_ atmosphere in a Galaxy 48R incubator (New Brunswick Scientific, Edison, NJ, USA). The cells were maintained in DMEM (Dulbecco’s Modified Eagle’s Medium) supplemented with 10% fetal bovine serum. Cisplatin and carboplatin were used as reference drugs.

Trypsinized cells were plated in a 96-well plate at a density of 1·10^4^ cells per well and incubated for 24 h at 37 °C under a humidified atmosphere containing 5% CO_2_. The growth media were then removed and replaced with 200 μL of fresh media containing test compounds of varying concentrations. No precipitation was observed at all studied concentrations. After 24 h of incubation, the culture medium containing the compounds was removed, and a solution of nitrotetrazolium blue chloride (NBT) in DMEM (200 µL, 1 mg·mL^−1^) was added to the cells. Upon incubation for 1 h at 37 °C, 5% CO_2_, the DMEM/NBT solutions were aspirated, and the purple formazan crystals were solved in 200 μL of DMSO. The absorbance of each well was quantified at 540 nm using a Tecan Spark 10M microplate reader. Absorbance values were normalized to the untreated wells and plotted as concentration of compounds versus percentage of viability. The resulting dose–dependent curves were analyzed using a logistic sigmoid function. The results of the experiments were presented as the average of three independent experiments, each carried out with four replicates per concentration level. The values of 50% growth inhibitory concentration (IC_50_) and maximal inhibition were calculated in comparison with the untreated controls. Stock solutions of the complexes were freshly prepared in DMSO and diluted with the medium (maintaining a final DMSO concentration in the medium not exceeding 1%). Stock solutions of the ligands were prepared in ethanol prior to serial dilution.

### 4.6. Molecular Docking Calculations

Docking studies were carried out on two organic ligands, two solvent-free complexes in trans-configuration, and two solvent-free complexes in cis-configuration. Prior to use, the structures of the compounds were taken from X-ray crystallography (except for metal complexes in cis-configuration) data and optimized using the quantum chemistry program ORCA 4.2.1 [88] using B3LYP/6-311++G(3df, 3pd) for copper and B3LYP/6-31G(d) for non-metallic atoms.

The calculations of the binding orientations of the studied compounds in active sites were performed using GOLD Version 2020.1 (CCDC, UK) [89]. The GoldScore and ChemScore scoring functions were applied to predict binding poses and relative energies. The parameter file for GOLD was adapted to include parameters of copper atoms which are not included in GOLD’s database.

For docking studies, the structures of R2 RNR (PDB ID: 1W68), Hsp70 (PDB ID: 1S3X), and Hsp90 (PDB ID: 1YES) were used. The centers of the binding pockets were defined (Ser264 for R2 RNR, Asn51 for Hsp90, Arg272 for Hsp70) with 10 Å radii. The images were generated using UCSF Chimera Software [90].

## Data Availability

The data are available from the corresponding authors upon reasonable request.

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
