# Peer review of "Unsymmetrical Trifluoromethyl Methoxyphenyl β-Diketones: Effect of the Position of Methoxy Group and Coordination at Cu(II) on Biological Activity"

_molecules, 2021, doi:10.3390/molecules26216466_

Round 1

Reviewer 1 Report

The authors nicely addressed concerns regarding complex stability in simulated biological fluid. That is a worthy addition to the manuscript. In addition, they added data points to their dose-response plot in Hela cells to ensure it demonstrates sigmoidal character. However, they did not extend the concentration range in their Vero cell dose-response plot to get a true CC50 value. In addition, the SI between the Vero and HeLa cells looks alarmingly low, thereby making these poor chemotherapeutic leads. Similarly, the selectivity index is too low for the compounds to serve as safe, systemic antibiotics. The authors also failed to test an established positive control in their MIC determinations, making the data suspect. As described previously, the docking studies are meaningless without experimentally validated biological targets and inclusion of virtual positive and negative controls. That key issue was not addressed by the authors in this revision.

I appreciate the authors’ effort in establishing compound stability in SBF; however, these very low potency compounds with relatively high cytotoxicity are unlikely to be of interest to the med chem community. The docking studies are an interesting inclusion but are likely irrelevant without experimental data to support the choice of virtual targets. Therefore, I do not find this work worthy of publication in a high-impact journal like Molecules

Author Response

Comment:

The authors nicely addressed concerns regarding complex stability in simulated biological fluid. That is a worthy addition to the manuscript. In addition, they added data points to their dose-response plot in Hela cells to ensure it demonstrates sigmoidal character. However, they did not extend the concentration range in their Vero cell dose-response plot to get a true CC50 value. In addition, the SI between the Vero and HeLa cells looks alarmingly low, thereby making these poor chemotherapeutic leads. Similarly, the selectivity index is too low for the compounds to serve as safe, systemic antibiotics.

Reply:

We did not extend the concentration range for Vero cell dose-response plots, because this cell line is currently not available for experiments in our research group. For current studies, we are using another cell line (LB929). Moreover, we believe that additional experiments with increased concentrations towards Vero would be of poor scientific value since such concentrations would be too high for treatment of both cancer and normal cells. Finally, the obtained curve shape is sufficient to allow definite conclusions about activity of complexes against normal cells which would not be affected by the extended data. At concentrations ranging close to MIC values towards S. aureus, approximately 50% of the Vero cells were still alive. We believe that these results will be useful for further metal-based drug development.

Comment:

The authors also failed to test an established positive control in their MIC determinations, making the data suspect. As described previously, the docking studies are meaningless without experimentally validated biological targets and inclusion of virtual positive and negative controls. That key issue was not addressed by the authors in this revision.

I appreciate the authors’ effort in establishing compound stability in SBF; however, these very low potency compounds with relatively high cytotoxicity are unlikely to be of interest to the med chem community. The docking studies are an interesting inclusion but are likely irrelevant without experimental data to support the choice of virtual targets. Therefore, I do not find this work worthy of publication in a high-impact journal like Molecules.

Reply:

We did not fail to test an established positive control in the MIC determinations. A series of commercial antibacterial agents including ampicillin, tetracycline, kanamycin (10 mg, Research Center of Pharmacotherapy, Saint Petersburg, Russia) was used as references in preliminary experiments using the disk diffusion method. The data revealed that the tested compounds are effective against the microorganisms with comparable activity (some data are preserved in photo, Fig. S4 was added). We made revisions of the manuscript (sections 2.4 and 4.4) and supplementary. Beyond all doubt, we will include the appropriate positive controls into all our subsequent biological experiments. The very strains are out of access for us now. Nevertheless, we believe that the data are robust and the conclusions on activity of our complex towards S. aureus ATCC 25923 are valid.

The objective of docking studies was comparative estimation of affinity for the reasonable selected targets. The results are intended for making a hypothesis regarding the origin of the observed biological effects.

We would like to extend a heartfelt “thank you” for the reviewers and editor, as the comments will be of great help for our work.

Reviewer 2 Report

The authors make all requested corrections. The paper could be publish in this form.

Author Response

Thank you very much

Reviewer 3 Report

In general, the paper entitled „ Unsymmetrical trifluoromethyl methoxyphenyl β-diketones: effect of the position of methoxy group and coordination at Cu(II) on the biological activity” (molecules- 1387532) is in the scope of Molecules. I find the methods used reliable. In my opinion, the presented data are novel as they are concerned on the quite new type of drugs, i.e. with dual anticancer and antimicrobial activities. Although the IC50 concentrations of tested complexes of Cu(II) and β-diketones derivatives against cancer (HeLa) and normal (Vero) cells are quite similar, the complexes seemed to be a promising pharmacophores to create new, bioactive drugs. The Authors concluded that different biological activities of parent ligands and complexes can result from their bioavailability which in turn can be connected with their structures. Hence, I find as reasonable to conduct the in silico studies on relationship between the structure and pharmacokinetic/drug-likeness properties of tested compounds. Moreover, as it was tested the cytotoxicity of compounds towards high-density cell cultures of cancer and normal cells, I find as more appropriate to use CC50 (50% cytotoxicity concentration) than IC50 (50% inhibitory concentration, more suitable to proliferation inhibition).

Author Response

We appreciate the referee’s comments and suggestions. We will take them into account for our further studies.

Round 2

Reviewer 1 Report

I appreciate that some materials are unavailable to the authors to conduct additional (or revised) recommended experiments, but the issue remains that several experiments were executed without positive or negative controls--inclusions that are part of the "scientific method" to validate procedures and systems. In addition, the indicated compounds have such low selectivity indices (i.e., they have toxicity comparable to their potency) that these would not be regarded as 'hits' or 'leads' by most researchers. If the authors could enhance selectivity for cancer cells or microbes relative to healthy cells, then I would be inclined to recommend for publication. However, in my view, the lack of experimental controls, particularly in the cell viability assays, coupled with relatively high toxicity does not befit publication in higher impact journals such as Molecules.

This manuscript is a resubmission of an earlier submission. The following is a list of the peer review reports and author responses from that submission.

Round 1

Reviewer 1 Report

The authors report the preparation, characterization (including X-ray data), and limited in vitro cell-based biological activities of known (and commercial) ligands HL0, HL1 and their Cu complex 3 . From a structural inorganic chemistry perspective, this might be publishable work due to the acquired X-ray data; however, from a medicinal chemistry perspective, which is the pretext of the manuscript, every study is fundamentally flawed. Here is listed a few of the many serious problems associated with the investigation:

  • There is no evidence the complexes remain intact in biological fluids. The first study conducted should have been related to the stability of the complexes with respect to time in simulated biological fluid (including appropriate ions) + albumin at 37 C. If the complexes decompose, then there is no point in their further evaluation in any biological context.
  • The dose-response studies with Vero And HeLa cells are flawed. The authors did not evaluate the complexes across broad enough concentration ranges to get meaningful data. In fact, the “curve” is effectively linear in the HeLa cell plot. This may reflect concentration-dependent non-specific interactions (e.g., compound aggregates, cell co-factor chelation, ROS generation in the reaction media, etc.) disrupting general target cell protein stability or activity and/or too narrow of a drug concentration range to elicit a sigmoidal response. The former may be true, and the latter is unequivocally true. Was poor solubility a limiter in the concentration range evaluated?
  • The potency data from the (flawed) HeLa cell cytotoxicity study reflects that of nothing more than a potential very weak “hit” (IC50 >117 uM), and one might only consider the complex as a very weak hit after accounting for compound solution stability and solubility—curiously, neither of which was reported.
  • The comparative data between the Vero cell and the HeLa assays (Figure 4) suggests a very poor selectivity index (SI < 2). Such unacceptably high relative toxicity may be associated with non-specific protein interactions as indicated above. This is relative toxicity/poor selectivity is not acceptable for any potential lead compound.
  • The high MIC values (Table 1) are not encouraging, particularly given the potential toxicity reflected in the Vero cell assays. Again, the SI is just too low to consider these complexes further or to generate any interest.
  • The docking study reported in section 2.5 is meaningless. The only biological assay data provided was from cell-based phenotypic screens, yet the authors conveniently selected three “possible” protein targets (of many hundreds of potential targets—or even more if non-specific interference is the cause of the noted weak cytotoxicity) for docking. Even then, they inexplicably used no positive and negative controls for comparison in the docking study, and they only considered relative binding energy differences (in the absence of waters, mind you) using a rigid docking model with a small series of very similar compounds (Table 2). Since the binding energy scores are relative, there is nothing to compare the scores against except the few compounds in the same series. Thus, the only conclusion that can be drawn from this docking study is: one compound in the series is slightly better than the others at binding in a particular pose within specifically selected locations in specifically (and rather arbitrarily) selected proteins in specifically selected protein conformations in the absence of competing waters. In other words, there is nothing of value that can be determined from the docking study the way in which it was conducted, and it states absolutely nothing about plausible drug targets or plausible mechanisms of action based upon correlations with any ascertained biological data. Basically, this is no better than data one might obtain from an initial structure-based docking study involving only six compounds in a single pose interacting in three protein binding sites (i.e., R2 RNR, Hsp70 and Hsp90), assuming those three proteins have no known inhibitors (which, of course, is not the case) to serve as positive controls in the docking model. Hence, all of page 11 is conjectural, mostly irrelevant, and does not belong in a science manuscript.

There are many other problems with the language used and the conclusions drawn that would have to be remedied prior to consideration for publication. However, the scientific method (outside of the preparation and X-ray analyses of complexes 1 and 3, which may be publication worthy in a completely different context) is terribly flawed and there is nothing of value in terms of biological activity or associated analysis provided herein. I strongly urge the authors to recast this from a purely structural inorganic chemistry perspective (much as they did with closely related complexes in reference 40) to salvage what they can for publication. However, I cannot recommend this study be published in Molecules or any other journal in the context of medicinal chemistry.

Author Response

Referee 1

Comment:

There is no evidence the complexes remain intact in biological fluids. The first study conducted should have been related to the stability of the complexes with respect to time in simulated biological fluid (including appropriate ions) + albumin at 37 C. If the complexes decompose, then there is no point in their further evaluation in any biological context.

Reply:

Copper diketonates are known to be pretty stable. In accordance with our previous report (Khamidullina et al. J. Mol. Struct. 2019, 1176, 515–528, doi:10.1016/j.molstruc.2018.08.112), ligand-free copper(II) does not exhibit antimicrobial activity under the mentioned conditions in contrast to the studied complexes. In both this and previous works, the complexes are reported to have lower MICs than the initial ligands. If the complexes would decompose then MICs of the complexes and the ligands would be equal.

Moreover, if complexes would decompose then their cytotoxicity would be caused by non-specific action of copper ions on cells. In this case, cytotoxicity of the complexes towards HeLa cell line would be the same. We demonstrated that the activities were different (Figure 4, Table S7). The ligand HL0 and the corresponding complex cis-[Cu(L0)2(DMSO)2] exhibited similar low maximal inhibition values for Vero line, but differed significantly for HeLa line (33 % at 536 µM for HL0, 69 % at 138 µM for cis-[Cu(L0)2(DMSO)2]). Therefore, the complexes do not decompose in Dulbecco’s Modified Eagle’s Medium that can be used as an alternative to conventional SBF (Lee et al. Acta Biomaterialia. 2011, 7, 2615–2622, doi: 10.1016/j.actbio.2011.02.034).

Finally, prior to any investigations of properties we perform stability tests by the UV-vis spectroscopy, which showed that the complexes concerned were stable in solutions over two weeks at least.

Comment:

The dose-response studies with Vero and HeLa cells are flawed. The authors did not evaluate the complexes across broad enough concentration ranges to get meaningful data. In fact, the “curve” is effectively linear in the HeLa cell plot. This may reflect concentration-dependent non-specific interactions (e.g., compound aggregates, cell co-factor chelation, ROS generation in the reaction media, etc.) disrupting general target cell protein stability or activity and/or too narrow of a drug concentration range to elicit a sigmoidal response. The former may be true, and the latter is unequivocally true. Was poor solubility a limiter in the concentration range evaluated?

Reply:

Indeed, poor solubility is a limiter in the concentration range evaluated and drug concentration range is quite narrow. However, the plot is not linear, it is a truncated sigmoid, the data unambiguously demonstrate the differences in activity. If possible non-specific interactions would take place, the trend would persist on both normal and tumorigenic cells. The data obtained in HeLa cells in fact clearly show the differences between the complexes, although the effect on Vero cells is almost the same.

Comment:

The potency data from the (flawed) HeLa cell cytotoxicity study reflects that of nothing more than a potential very weak “hit” (IC50 >117 uM), and one might only consider the complex as a very weak hit after accounting for compound solution stability and solubility—curiously, neither of which was reported.

Reply:

IC50 for one of the complexes is 88.3±12.1 μM: “The cis-[Cu(L0)2(DMSO)2] showed higher activity against HeLa cells (IC50 = 88.3±12.1 μM) than the cis-[Cu(L1)2(DMSO)] (IC50 = 117.1±15.1 μM)”. Despite the activity is moderate, we believe that it should be discussed in the manuscript. We suppose that the limitations are not due to the poor solubility but are rather caused by several factors resulted in limited bioavailability. The discussed results may be useful for understanding mode(s) of action of such compounds on cells.

In addition, the compounds have adequate solubility under the used conditions with no precipitation from both stock solutions and nutrient/culture media. Thus, we assume that solubility issue is not worth mentioning in the manuscript.

Comment:

The comparative data between the Vero cell and the HeLa assays (Figure 4) suggests a very poor selectivity index (SI < 2). Such unacceptably high relative toxicity may be associated with non-specific protein interactions as indicated above. This is relative toxicity/poor selectivity is not acceptable for any potential lead compound.

Reply:

We do not claim our complexes as pharmaceutical products suitable for internal use.

Copper complexes can demonstrate interesting in vivo effects on crucial processes (cancer proliferation, angiogenesis and metastasis), since cancerous tissues are more sensitive to copper and have a greater demand for it than normal cells for proliferation and survival. Effects and benefits depend on drug delivery method.

Considering the exponential growth of bacterial resistance, any information on the activity of new substances will be useful for further drug search and development.

Comment:

The high MIC values (Table 1) are not encouraging, particularly given the potential toxicity reflected in the Vero cell assays. Again, the SI is just too low to consider these complexes further or to generate any interest.

Reply:

At concentrations ranging from 100 to 200 μM, a complete growth inhibition of the sensitive bacteria was observed, but about 50% of Vero cells were still remain viable. The data are noteworthy.

Comment:

The docking study reported in section 2.5 is meaningless. The only biological assay data provided was from cell-based phenotypic screens, yet the authors conveniently selected three “possible” protein targets (of many hundreds of potential targets—or even more if non-specific interference is the cause of the noted weak cytotoxicity) for docking. Even then, they inexplicably used no positive and negative controls for comparison in the docking study, and they only considered relative binding energy differences (in the absence of waters, mind you) using a rigid docking model with a small series of very similar compounds (Table 2). Since the binding energy scores are relative, there is nothing to compare the scores against except the few compounds in the same series. Thus, the only conclusion that can be drawn from this docking study is: one compound in the series is slightly better than the others at binding in a particular pose within specifically selected locations in specifically (and rather arbitrarily) selected proteins in specifically selected protein conformations in the absence of competing waters. In other words, there is nothing of value that can be determined from the docking study the way in which it was conducted, and it states absolutely nothing about plausible drug targets or plausible mechanisms of action based upon correlations with any ascertained biological data. Basically, this is no better than data one might obtain from an initial structure-based docking study involving only six compounds in a single pose interacting in three protein binding sites (i.e., R2 RNR, Hsp70 and Hsp90), assuming those three proteins have no known inhibitors (which, of course, is not the case) to serve as positive controls in the docking model. Hence, all of page 11 is conjectural, mostly irrelevant, and does not belong in a science manuscript.

Reply:

The objectives of our research included neither the consideration of commercial drugs or known inhibitors nor comparison of the tested compounds with them.

The tasks of the study were to check whether the complexes have a higher affinity for the selected targets in comparison with free ligands and with one another. With regard to the choice of targets, the rationale is given at the beginning of Section 2.5. These are not randomly chosen targets. However, we will include other targets in our next works. We appreciate the detailed comment.

Reviewer 2 Report

see attachment

Author Response

Comment:

The authors affiliation should be revised because the last two authors have unidentified affiliations:

Reply:

Corrected

Comment:

There is no reference for IR spectra.

Reply:

The references have been added

Comment:

Sometimes appears и instead of and (page 4, row 16; page 5, row 209) Page 4, row 169: pyramidal [42] instead of pyramidal[42].

Reply:

Corrected

Comment:

Page 6, row 239: …we performed it earlier) [42,44–46]. instead of we performed it earlier).[42,44–46]

Reply:

Corrected

Comment:

At caption 2.4. Antimicrobial activity all strains should be typed using italic letters (e.g. S. aureus, E. coli) (page 7, rows 267-269; page 8, rows 284, 289, 298).

Reply:

Corrected

Reviewer 3 Report

In this manuscript, Copper(II) complexes with 1,1,1-trifluoro-4-(4-methoxyphenyl)butan-2,4-dione (HL1) were synthesized and characterized by elemental analysis, FT-IR spectroscopy, and single crystal X-ray diffraction. According to what the authurs disclosed, the copper(II) complexes with the trifluoromethyl methoxyphenyl-substituted β-diketones could be considered as promising anticancer agents with antibacterial properties. I am not an expert in the field of study on bioactivities of chemicals.  But the compounds reported here seems not behave as very potential anti-cancer drugs. The authors should modify the sturcture of the ligand on the metal and test more complexes to get better result. I would like to say this work might be too eary to be published, since the synthesis of these complexes are simple.

Author Response

Comment:

In this manuscript, Copper(II) complexes with 1,1,1-trifluoro-4-(4-methoxyphenyl)butan-2,4-dione (HL1) were synthesized and characterized by elemental analysis, FT-IR spectroscopy, and single crystal X-ray diffraction. According to what the authurs disclosed, the copper(II) complexes with the trifluoromethyl methoxyphenyl-substituted β-diketones could be considered as promising anticancer agents with antibacterial properties. I am not an expert in the field of study on bioactivities of chemicals.  But the compounds reported here seems not behave as very potential anti-cancer drugs. The authors should modify the sturcture of the ligand on the metal and test more complexes to get better result. I would like to say this work might be too eary to be published, since the synthesis of these complexes are simple.

Reply:

We believe that the simplicity or complexity of compounds is not the main criterion for publication. Small molecules with a simple structure are widespread in medicinal chemistry, and many lead molecules have a simple structure. Therefore, the simplicity of structure/synthesis is rather an advantage than a disadvantage. Examples of such small molecules are cisplatin, carboplatin and other platinum complexes, as well as the derivatives of diketones (e.g. thenoyltrifluoroacetone, Casiopeínas copper complexes), etc.

Reviewer 4 Report

The study centered on the synthesis and characterization of Copper(II) complexes with 1,1,1-trifluoro-4-(4-methoxyphenyl)butan-2,4-dione (HL1). In particular, the authors have performed an elemental analysis, followed by an analysis of the FT-IR spectroscopy, and single crystal X-ray diffraction. Among other interesting findings, they have also examined the effect of changing the position of methoxy group at aromatic ring in the ligand moiety of complexes on their antimicrobial and cytotoxic activity. Finally, it was shown that complex 3 has lower bioavailability and reduced bioactivity than were expected due to strong intermolecular contacts. Although this work would be interesting to the readers of the journal, I have some minor suggestions to the authors, which should consider and revise their paper. They are given below.

1- What kind specific intermolecular interactions feasible in the chemical systems examined? Why were they not shown in various molecular structures provided? 

2- Did Cu favor only planar structure Figure 5? 

3- Why were hydrogen atoms missing in all structures? 

4- Coordination chemistry of Cu is not detailed. Further fundamental discussion is necessary 

5- Why not the results of the DFT calculations performed rigorously discussed to validate the chemical bonding topologies more accurately? 

6- The paper lacks an appropriate conclusion section and hence is not suitable for publication in its current form. 

7- Background references are not up to the mark and writing standard must be significantly improved. 

Author Response

Comment:

What kind specific intermolecular interactions feasible in the chemical systems examined? Why were they not shown in various molecular structures provided?

Reply:

The discussion of specific intermolecular interactions has been added in the section 2.2. Structural studies.

Comment:

Did Cu favor only planar structure Figure 5?

Reply:

According to DFT, for all structures Cu favors distorted planar structure.

Comment:

Why were hydrogen atoms missing in all structures?

Reply:

For the sake of clarity, in the structures of copper complexes hydrogen atoms are omitted.

Comment:

Coordination chemistry of Cu is not detailed. Further fundamental discussion is necessary

Reply:

Please see section 2.2. “Structural studies” having the detailed discussion of Cu coordination chemistry.

Comment:

Why not the results of the DFT calculations performed rigorously discussed to validate the chemical bonding topologies more accurately?

Reply:

The DFT calculations were used only for simulation of the structures of the compounds for docking studies in cases where X-ray data were absent.

Comment:

The paper lacks an appropriate conclusion section and hence is not suitable for publication in its current form.

Reply:

Please see section 3 which includes the conclusions.

Comment:

Background references are not up to the mark and writing standard must be significantly improved.

Reply:

Writing standard was improved.

Round 2

Reviewer 1 Report

The revised manuscript remains unsuitable for publication because the methods, interpretations, and outcomes are poorly designed or deceptive. None of this is improved based upon the minor corrections introduced by the authors in their revised manuscript.

Here are the facts:

The authors prepared and carefully characterized new Cu-complexes that are minor derivatives of those previously published by the same authors. Perhaps that part is publishable, but likely as a letter in a specialty, low-impact journal--not in Molecules.

However, major issues make the rest of the study unsuitable for publication:

  1. The stabilities of the Cu complexes were not determined in a simulated biological fluid at body temperature. The complexes might not remain intact in such environments.

     2.The Cu complexes are nearly as cytotoxic to healthy (Vero) cells as they are to cancer (HeLa) cells and bacteria. An acceptable selectivity index for a “hit” is SI >10; here SI <2!

  1. The IC50 values are very high and are 4-5 times above the accepted minimum threshold of potency for a “hit”. It is highly unlikely anyone would have any interest in these complexes due to their very low potency and very high relative toxicity.
  2. The bioassays were not validated with controls (i.e., the authors’ failed to follow the scientific method) and were truncated, thereby leading to potentially erroneous IC50 values. In fact, the Vero cell study was halted at 50% cell viability. That is unacceptable and likely leads to erroneously favorable potency/safety data.
  3. Solubility is a vital parameter for any biologically active compound. The authors note in their rebuttal the dose-response plots were truncated due to limited Cu-complex solubility, yet they do not state this key observation in the manuscript and ignore it in the discussion of their poorly-constructed dose-response studies.
  4. Just as the authors used no controls in their bioassays, they used no controls (e.g., comparison with known inhibitors associated with X-ray co-crystal structures of the interrogated proteins) in their docking studies. Hence, the docking scores are virtually meaningless.
  5. The Vero cell toxicity and the linear response in the bioassay of cis-[Cu(L0)2(DMSO)2] suggest the complexes are not associating with a single target nor even a couple of targets, but rather are indiscriminately damaging cellular components. Hence, docking studies involving three arbitrarily selected proteins with which the complexes might associate is meaningless until target-based dose-responses are validated in suitable assays with appropriate controls. The reported docking study is deceptive to the unknowing reader.

This whole study is “smoke and mirrors”. Investigations of this type do not belong in high-quality journals like Molecules, nor in any journal, for that matter. This manuscript should not be considered further for publication.

Reviewer 3 Report

Synthetically, I do not see significant contribution of this work. The compounds reported here seems not behave as very potential anti-cancer drugs. I strongly suggest the authors to modify the sturcture of the ligand to get a really potential lead compound. I would like to see the electronic and steric effects of the substituents on the biological activity. These structrue-activity relationship is important for the design of biological active compounds. Otherwise, I do not think it is enough to be published.

Reviewer 4 Report

Authors of the work have considered all my suggestions and have revised their paper. As it seems, the revised version of the paper is nearly complete, and hence may be considered for possible publication